# Mixed methods implementation research of cognitive stimulation therapy (CST) for dementia in low and middle-income countries: study protocol for Brazil, India and Tanzania (CST-International)

Aimee Spector,[1] Charlotte R Stoner,[1] Mina Chandra,[2] Sridhar Vaitheswaran,[3] Bharath Du,[4] Adelina Comas-Herrera,[5] Catherine Dotchin,[6,7] Cleusa Ferri,[8] Martin Knapp,[5] Murali Krishna,[4] Jerson Laks,[9] Susan Michie,[1] Daniel C Mograbi,[10,11] Martin William Orrell,[12] Stella-Maria Paddick,[13] Shaji KS,[14] Thara Rangawsamy,[15] Richard Walker[6,7]

For numbered affiliations see end of article.

**Correspondence to**
Dr Charlotte R Stoner;
c.stoner@ucl.ac.uk

## ABSTRACT

**Introduction** In low/middle-income countries (LMICs), the prevalence of people diagnosed with dementia is expected to increase substantially and treatment options are limited, with acetylcholinesterase inhibitors not used as frequently as in high-income countries (HICs). Cognitive stimulation therapy (CST) is a group-based, brief, non-pharmacological intervention for people with dementia that significantly improves cognition and quality of life in clinical trials and is cost-effective in HIC. However, its implementation in other countries is less researched. This protocol describes CST-International; an implementation research study of CST. The aim of this research is to develop, test, refine and disseminate implementation strategies for CST for people with mild to moderate dementia in three LMICs: Brazil (upper middle-income), India (lower middle-income) and Tanzania (low-income).

**Methods and analysis** Four overlapping phases: (1) exploration of barriers to implementation in each country using meetings with stakeholders, including clinicians, policymakers, people with dementia and their families; (2) development of implementation plans for each country; (3) evaluation of implementation plans using a study of CST in each country (n=50, total n=150). Outcomes will include adherence, attendance, acceptability and attrition, agreed parameters of success, outcomes (cognition, quality of life, activities of daily living) and cost/affordability; (4) refinement and dissemination of implementation strategies, enabling ongoing pathways to practice which address barriers and facilitators to implementation.

**Ethics and dissemination** Ethical approval has been granted for each country. There are no documented adverse effects associated with CST and data held will be in accordance with relevant legislation. Train the trainer models will be developed to increase CST provision in each country and policymakers/governmental bodies will be continually engaged with to aid successful implementation. Findings will be disseminated at

### Strengths and limitations of this study

► Utilisation of the consolidated framework for implementation research in planning this research will facilitate subsequent adaptation and generalisability for other programmes implementing evidence-based therapies in low/middle-income countries (LMICs).

► Collaborative investigation with patient and public involvement representatives, experts in the field and policymakers in each country to identify the unique country or site-specific barriers and facilitators and possible implementation strategies to compensate for these will help ensure sustainable implementation of cognitive stimulation therapy (CST).

► Valid and reliable outcome measures for each country will be used, countering the cross-cultural limitations of some existing measures and projects.

► Mixed methodology will ensure extensive evaluation of both objective success of implementation and the subjective experience of delivering and engaging with CST in each country, which will inform implementation.

► Analysis of the cost of CST delivery and any financial benefits of CST provision by comparing the cost of supporting people with dementia pre-CST and post-CST in each country will inform the sustainability of CST in LMICs.

conferences, in peer-reviewed articles and newsletters, in collaboration with Alzheimer's Disease International, and via ongoing engagement with key policymakers.

## INTRODUCTION

Dementia is a substantial global challenge affecting around 46.8 million people globally,

with an estimated annual worldwide cost of 818 billion US dollars.[1] Currently, there are 27.3 million people with dementia in low/middle-income countries (LMICs). Between 2015 and 2050, the number of older people living in high-income countries (HICs) is forecast to increase by 56%, compared with 138% in upper middle-income countries, 185% in lower middle-income countries and 239% in low-income countries. There is a major disjunction between global distribution of dementia prevalence, with 58% of cases currently living in LMICs, and costs, with 87% of globally spending on dementia incurred in HIC. In response, the WHO stated that: 'A sustained global effort is thus required to promote action on dementia and address the challenges posed… No single country, sector or organisation can tackle this alone'[2] (p42).

Despite growing numbers of people living with dementia, service provision remains limited in many world regions. For example, >1.5 million people in Brazil may have dementia but only 20% have been diagnosed.[3] While the current government policy of providing free high cost medication has benefitted many people with dementia, dementia awareness remains limited and no formal and validated psychological or social interventions are currently offered. In India, around 4.4 million people have dementia and this is expected to increase to over 10 million by 2040.[1] Dementia has traditionally been considered to be a part of normal ageing and not a medical problem. This has often resulted in delayed help seeking, with an estimated treatment gap of over 90%. Only around 5% of those with dementia receive a formal diagnosis and appropriate treatment, due to a lack of trained professionals.[4] The only dementia prevalence study conducted in Tanzania identified an age adjusted prevalence of 6.4% in people aged 70 years and over, similar to prevalence in many HICs.[5] This equates to around 200 000 living with dementia nationally, almost all of whom will receive no specialist care. While some people will seek treatment from traditional healers or alternatives, dementia can be associated with high levels of comorbidity, mortality and high carer burden.[6]

In countries where access to medication can be difficult, non-pharmacological interventions may be an effective means of treating people with dementia.[7] However, the evidence base for some non-pharmacological interventions is variable and, as they are predominantly developed in HIC,[8] they may be less suitable for use in LMICs.

Cognitive stimulation therapy (CST) is a brief, evidence-based, effective and cost-effective in the UK intervention for people with mild to moderate dementia. Developed in the UK, it involves 14 sessions over 7 weeks. The aim of CST is to improve cognitive function through themed group activities, which implicitly stimulate skills including memory, executive function and language through tasks such as categorisation, word association and discussion of current affairs. A Cochrane systematic review of 15 randomised controlled trials (RCTs) found consistent evidence that CST benefits cognition in mild to moderate dementia, over and above any medication effects.[9]

Furthermore, CST demonstrates the best evidence for improving cognitive functioning among all psychosocial interventions.[10]

In the first RCT,[11] significant improvements in both cognition and quality of life of CST compared with usual care (n=201) were documented. A 'numbers needed to treat' analysis found that improvements in cognition were similar to those following use of anticholinesterase inhibitors. An economic analysis, in which the cost of running CST groups in addition to the differences in use of services between the treatment and control groups were calculated and analysed alongside evidence on cognitive and quality of life benefits, found CST to be cost-effective.[12] In 2006, the UK National Institute of Clinical Excellence guidance on dementia[13] recommended 'structured group Cognitive Stimulation, irrespective of any anti-dementia drug prescribed'. It remained the only non-pharmacological intervention recommended to improve cognition, independence and well-being in the updated 2018 guidelines.[14] A report by the NHS Institute of Innovations and Improvements indicated that if CST were widely implemented, it would save the National Health Service (NHS) £54.9 million annually through combining healthcare cost savings with quality of life improvements.[15]

Global adaptation and evaluation of CST coincided with the World Alzheimer's Report 2011,[16] which stated that CST should routinely be given to people with early stage dementia and advocated CST as an effective, low-cost intervention in developing countries. To ensure that CST provision is translated into routine clinical practice, implementation research that addresses practical issues such as referral pathways and barriers to successful provision is needed. The aim of the CST-International research programme is to develop, test, refine and disseminate implementation strategies for CST for people with dementia in three diverse parts of the world. The primary objective is to create a sustainable CST implementation programme that enhances quality of life and cognition for people with dementia. A secondary objective is to increase awareness and skills in the detection and management of dementia, both for health workers and families.

One upper middle-income (Brazil), one lower-middle income (India) and one low-income (Tanzania) country have been selected. They have all: (1) begun or completed feasibility or pilot work on CST with positive results, with two sites previously awarded funding for feasibility work; (2) previously translated and adapted the CST manual, following the same recommended process[17] and (3) engaged local stakeholders and gathered initial data on implementation.

## METHODS AND ANALYSIS

The CST-International team is comprised of both national and international staff. The international team is co-ordinated in the UK and consists of psychologists, psychiatrists and researchers at higher education institutions and NHS trusts. National teams in each country

are comprised of psychologists, psychiatrists, physicians, nurses, occupational therapists and researchers. National teams belong to higher education institutions, non-governmental organisations, government healthcare facilities, public and private healthcare facilities.

The methods used was informed by theories and frameworks, where the focus is on improved translation of interventions to routine practice. This enables interventions to be sustainable and continue to be used following the completion of research trials. The consolidated framework for implementation research (CFIR)[18] is an amalgamation of 19 models of implementation and incorporates various theories around innovation, organisational change, implementation, knowledge translation, research uptake and dissemination. The CFIR has five domains: (1) characteristics of the intervention, (2) the outer setting, (3) the inner setting, (4) characteristics of individuals and (5) process. These are associated with 39 further constructs such as the adaptability of the intervention, patient needs and resources, the structural characteristics of the intervention and knowledge and beliefs about the intervention.

### Phase I: exploring the barriers and facilitators to implementation (months 1–6)

The aim of phase I is to identify potential barriers and facilitators to implementation of CST in each unique context, identifying areas requiring further adaptation. Initially, this will be addressed using a systematic review of psychological and social intervention research in LMICs. The systematic review will be used to investigate the effectiveness of previous interventions and implementation barriers and facilitators.

### Stakeholder meetings

Phase I includes substantial patient and public involvement (PPI), in which the barriers and facilitators to implementation will be discussed in a series of stakeholder meetings with three diverse groups of stakeholders. Questions for each group were developed using the CFIR[18] framework and iterative draft questions were discussed in national and international teams before a final and standardised version was agreed on. Group 1 will involve policymakers and discussions will be facilitated around barriers to policy implementation. Specific issues to be addressed will include awareness of dementia and ways of managing it, perceived importance and complexity of the issue, available capital and capacity. Group 2 will consist of policy implementers who will be primarily responsible for facilitating CST groups in their own service or setting. This group will explore implementation issues around training and support required, accessibility of the manual in its current form, adaptations to the 1 day training model, how people will be recruited for CST groups and what degree of psychological and educational support and advertising might be needed. This group will discuss how stigma, cultural expectations, age and gender issues, perceived resistance from service users and transport barriers might be addressed. Group 3 will consist of people with dementia, carers and community leaders or village elders where appropriate. In this group, knowledge of dementia, stigma, gender issues and logistics issues such as transport and meeting spaces will be discussed. A minimum of 10 stakeholders per group will be used in each country and the research team in each country will be responsible for inviting stakeholderings to meetings using their existing connexions and exploring new ones. All stakeholder meetings will be preceded by introductory talks on both dementia and CST. This will ensure that all stakeholders have at least a minimum understanding of both. Stakeholders will be encouraged to give their opinions on the topics discussed and facilitators will ensure that all stakeholders understand the purpose and requirements of the meetings.

Example questions for each stakeholder group include:
► Group 1: what, if any, national guidelines regarding dementia treatments are available and what guidelines should CST be in at the end of this study?
► Group 2: what are the known barriers people encounter when accessing healthcare services generally?
► Group 3: what would make you/your friend or relative with dementia more likely to attend CST sessions?

### Developing 'implementation mechanisms'

Findings from phase I will be collated for each country and a descriptive analysis will be used to document both similarities and unique implementation issues across countries. National teams will tabulate all identified barriers and facilitators and, using the CFIR as a guide and in consultation with experts in each country, propose mechanisms that could be used to overcome each barrier or support each facilitator. These will be tabulated alongside the barriers and facilitators identified and proposed mechanisms will be rated according to how essential the mechanism is to support implementation and how difficult the proposed mechanism is to execute. In the first instance, ratings will be conducted by CST-International staff and investigators in their role as experts in their respective countries. Following these ratings, an advisory group consisting of a minimum of three representatives from each of the stakeholder groups will be asked to provide further ratings. The resulting matrix of ratings will be assigned numerical values and the mode used in order to weight mechanisms on how essential and easy they are to use. The results of this will be discussed in local teams where members will be asked to reach a consensus and justify which mechanisms are to be used.

### Phase II: development of implementation strategies (months 7–10)

The aims of phase II are to generate implementation plans ready to be tested in phase III, and to complete preparation work required, including adaptation to the training protocol and generating recruitment sources.

### Agreeing an 'implementation plan'

Data from phase I will be organised into country specific implementation plans and a local co-ordinator in each site will be responsible for writing this plan. The local co-ordinator will be a member of the national research team and be supervised by a site or country lead. Each plan will include written summaries of the implementation mechanisms to be used and the justification for doing so, with reference to available time and resources. The local co-ordinator will also agree action plans with individual investigators, in which specific mechanisms are assigned to members of staff to execute. Each plan will be circulated and approved by the key team (coapplicants, collaborators, advisory groups and PPI groups). A consensus meeting in London, UK, which lead coapplicants from each country will attend, will be used to refine implementation plans in each country.

### Additional implementation activities

To ensure customised, country specific implementation, other tasks will be undertaken. First, adaptations to the 1 day CST training course will be considered. Training modules include the biopsychosocial model of dementia, screening using outcome measures of cognition, mood and quality of life. Second, researchers will prepare a cascade model of training to be used in phases III and IV. Third, a 3 hour dementia awareness course for people with dementia, carers and members of the general public will be developed. This course will be based on the successful 10/66 carer training intervention,[19] previous research in each country[20 21] and the CST training course. Fourth, further supplementary support needed for implementation will be considered. For example, nurses previously conducted physical examinations including blood pressure checks in Tanzania and India to help normalise the process of attending CST sessions. Fifth, psychoeducation, dissemination and advertising routes will be considered, with the overall aim being to generate appropriate referral routes. Previously, this was accomplished in Tanzania through screening days organised by religious leaders and village elders. Finally, staff will identify the resources needed to support implementation, for example, identifying recruiting sites, facilitators and room space in order to run CST sessions.

### Phase III: testing the implementation strategy with a study of CST (months 11–28)

The aims of phase III are to evaluate the feasibility of implementation strategies for each country. This will be accomplished using a study of CST where the following outcomes will be assessed: (1) adherence, attendance, acceptability and attrition, (2) number of trained facilitators and number of groups run, (3) the effectiveness of CST on outcomes of cognition and quality of life and (4) costs and the affordability of the intervention.

During phase III, CST facilitators will be trained to deliver CST using materials developed in previous phases. A minimum of eight CST groups in each country will also be established, recruiting 50 people with dementia in each country. Thus, the total sample size will be 150 people with dementia. The sample size was calculated pragmatically, based on discussions with each team regarding their available time, resources and the sample size required to evaluate the success of implementation strategies. To maximise recruitment, we will ensure that recruitment strategies target a range of healthcare settings and services including both private and public systems where appropriate. For example, primary health clinics and privately run day centres. National teams will be responsible for contacting these sites, using the most appropriate means. This may be through formal letters or email and teams will meet managers or leaders to explain the study in detail prior to their recruitment.

### Screening and inclusion criteria

The second version of the Mental Health Gap action programme (WHO),[22] which has a module on dementia, will be used by doctors, nurses and health workers to generate a list of suspected cases of dementia in the community. In India and Brazil, identified cases will be screened using the community screening instrument for dementia CSI-D brief.[23] This contains seven cognitive items and six informant items, taking <5 min to administer. In Tanzania, this will be replaced by the IDEA cognitive screen[24]; a six-item cognitive screen, which includes items from the CSI-D and the consortium to establish a registry for Alzheimer's disease 10 word list. It also contains an added matchstick task in place of the shape copying task, which has been validated in Nigeria. The screen has better reliability and validity in Tanzania, with a sensitivity of 84.6% and specificity of 89.1%. All suspected cases will be checked against International Classification of Diseases, 10th revision (ICD-10) criteria[25] for dementia. All screeners have previously been trained to use these tools and screening will take place in the most appropriate venue for each country. For example, a recently established memory clinic will be used for screening in Tanzania. Screening for CST will continue until the specified sample size for each country is reached.

The inclusion criteria have been informed by previous research but have been adapted to meet local needs. To be included, participants must:

1. Meet the ICD-10 criteria for dementia, as assessed by a trained clinician.
2. Be rated as having mild to moderate dementia on the Clinical Dementia Rating Scale.[26]
3. Have sufficient hearing and vision to follow conversation/comment on visual material.
4. Have the ability to participate in a group for 1 hour.
5. Be willing and able to complete measures of cognition and quality of life.
6. Be willing and able to travel to a group.

### Outcomes

Sociodemographic information collected will include age, gender, dementia subtype (if known), level of

**Table 1** Phase III outcome measures

| Domain | Measure | Items | Rater | Details |
|---|---|---|---|---|
| Cognition | Alzheimer's Disease Assessment Scale—Cognitive Subscale (ADAS-Cog)[32] | 21 | Person living with dementia (PwD) | Internationally recognised measure that includes three subscales: language, memory/new learning and praxis. It has been extensively validated in a range of settings and been adapted for use in in Sub-Saharan Africa as the main cognitive outcome in the IDEA study.[33] This research is currently guiding adaptation work in India. |
| Quality of life | WHO Quality of Life-Bref (WHOQOL-BREF)[34] | 26 | PwD | Consists of four domains: physical, psychological, social and environmental. Internal consistency for all domains is acceptable ($\alpha=>0.7$) and the tool has been extensively validated across LMICs.[35] Raw scores will be recorded (4–20), as per WHOQOL-Bref protocol. |
| Activities of daily living (ADLs) | EASY-Care Independence Scale (EASY-Care)[36] | 18 | PwD | Developed from existing measures of ADLs, the measure uses a weighting system to measure dressing, bathing, housework, preparing meals and feeding. Total scores range from 0 to 100, with higher scores denoting greater degree of dependence. It has been validated in LMIC's, most recently in India where internal consistency was reported as excellent. |
| Burden | Zarit Burden Interview (ZBI)[37] | 22 | Caregiver | Rates the impact of a person's disabilities on the caregivers' life. Responses are rated from 0 to 4, with higher scores indicating greater burden. Internal consistency is excellent ($\alpha=0.92$), however, despite some validation in LMICs including India,[38] there is some evidence to suggest it is not cross-culturally valid. |
| | Dementia Caregiver Experience Scale (DemCarES)[39] | 17 | Caregiver | Due to potential issues with the ZBI, the DemCarES will also be utilised to assess caregiver burden. The CES was developed recently in India to account for the unusually low levels of burden documented by existing measures. Internal consistency has been found to be excellent ($\alpha=0.91$) and the measure will be translated according to best practice and piloted in each of the countries. |
| Cost-affordability | Client Services Receipt Inventory (CSRI)[40] | N/A | Caregiver | The CSRI is used to collect information on service utilisation, income, accommodation and other cost-related variables. It has five sections consisting of: background information, accommodation and living situation, employment history, earnings and benefits, a record of services used and information about unpaid carers. Country-specific CSRIs will be used or developed over the course of the project. |
| | Resource Utilisation in Dementia (RUD)[41] | N/A | Caregiver | The RUD is designed for the collection of data pertaining to formal and informal care resource use across different countries and care systems. It includes items on accommodation, time spent assisting with activities of daily living and time spent assisting with instrumental activities of daily living. |

LMICs, low/middle-income countries.

literacy and education, ethnicity, previous or lifetime occupation, family composition and caregiver availability and arrangements. The same instruments will be used across sites (table 1), to facilitate cross-cultural comparisons and statistical analyses. Outcomes measures will be administered by trained research assistants at baseline (week prior to commencing CST) and follow-up (week following completion of CST). The study of CST is primarily to assess the effectiveness of the implementation strategies developed and is not powered for statistically

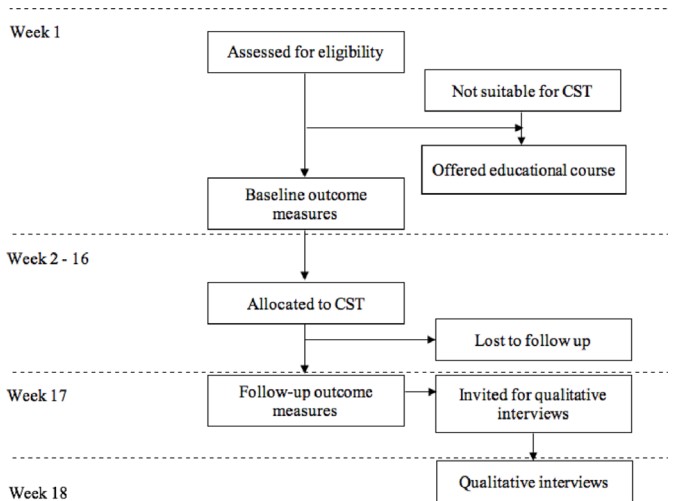

Week 1

Week 2 - 16

Week 17

Week 18

**Figure 1** Phase III participant flow. CST, cognitivestimulation therapy.

significant change. The main goal of the outcomes is to: (a) provide an indication of any improvements following CST and (b) demonstrate good practice through gathering outcome data. Outcome measures listed in table 1 have been translated, adapted and culturally validated for each setting. Adaptations to the Client Services Receipt Inventory and the Resource Utilisation in Dementia for each country are ongoing and will follow a standard operating procedure previously established. Any amendments to outcome measures will be informed by phases I and II of the project.

### Intervention

All participants who meet inclusion criteria and complete baseline assessments will receive 14 sessions of CST over 7 weeks (figure 1). The previously established and tested adaptions of the CST manual for each country will be followed. Examples of adapted activities include the 'current affairs' session in Tanzania including village news and events as there is limited awareness of national news. In Brazil, given its continental size; the orientation session will rely on local instead of national maps. In India, as older men would not traditionally be involved in cooking, the food session included budgeting for a meal. Task adaptation to accommodate illiteracy and uncorrected sensory impairment and use of locally available materials and equipment to ensure sustainability, will be essential in all places. Particularly in Tanzania and India, care will be needed to select a meeting place acceptable to all (eg, avoiding using a place of worship for groups of mixed religion). Two sessions may need to be held on the same day, to reduce travel time, with informal time before sessions to allow for traffic delays. Speicific venues for CST will be explored and recruited using our implementation strategies in phase II. However, we aim to recruit in two or more regions of each country, including Rio de Janeiro and Sao Paulo in Brazil; Chennai, Mysuru, Kerala and New Delhi in India; Kilimanjaro and Arusha in Tanzania. Groups will take place in both rural and urban settings which include

outpatient units, carers associations, Primary Health Care Centres and community settings.

A 3 hour family educational session will be offered by the researchers to all families of those who were screened and not included in the study, and to those who participate after completion of CST and follow-up outcome measures. The aim of this course will be to increase awareness of dementia in each country and will be developed based on findings in phase II. In addition, the course will contain information designed to combat common barriers across countries such as knowledge and awareness of both dementia and dementia treatments. Wherever possible, the course will be offered in a group format, both to encourage communication between family members and for the economy of time. People may be offered further support such as physical examinations, depending on the setting, needs and availability of resources.

### Qualitative interviews

In each country, qualitative interviews will be conducted after completion of CST, with equal numbers of service-users (people with dementia and their families), professionals training or facilitating CST and policymakers. Recruitment will continue until data saturation has been reached. Based on previous qualitative studies of CST, it is anticipated that ~15–20 interviews will be conducted in each country. Interview schedules will be developed and guided by the 39 items of the CFIR, with questions about feasibility, effects and implementation of CST directed to each stakeholder group. The aim of this is to further evaluate the implementation strategies utilised over the course of the grant including strengths, weaknesses, support needed and wider acceptability of the programme for the public, health professionals and politically. Narrative interviewing,[27] which focuses on lived experiences with minimal coaching or directing from the interviewer will be used by trained qualitative researchers in each site, who will also code and analyse results.

*Feasibility analysis* will consist of:
1. *Recruitment,* including people who were approached, agreed to attend, refused (and reasons for this) and met inclusion criteria.
2. *Attrition,* with details of numbers of those dropping out and reasons given for this. Twenty per cent (or less) attrition is generally considered acceptable.[28]
3. *Attendance,* average number of sessions and reasons for non-attendance.
4. *Acceptability of outcome measures,* examining whether completion was possible and missing data.
5. *Adverse events and side-effects,* routinely recorded according to ethical procedures.
6. *Adherence to manual.* This will enable us to assess whether people are delivering CST according to the protocol. People will be given a brief checklist to complete after each session, developed as part of the maintenance CST trial.[29]

*Agreed parameters of success for the study of CST* will consist of:

1. Number of people trained to deliver CST and number of people trained as CST trainers.
2. Number of groups run.
3. Total number receiving CST across settings and countries.

## Outcome measure and qualitative analysis

Analysis of outcome data will include descriptive (eg, mean, median, frequency) and inferential (eg, paired t-test, Wilcoxon signed-ranks test, percentage change) statistics using the Statistical Package for Social Sciences. Analysis of variance analyses with time as a within-subjects factor and group (CST vs treatment as usual) as a between-subjects factor will also be used, where appropriate . Data will be primarily presented for each site separately, with between-site comparisons made as appropriate. Data will be combined for analysis where this is justified, and it is meaningful to do so. Qualitative interviews will be audio recorded, transcribed, translated and analysed manually using interpretive phenomenological analysis,[30] which has been adapted and validated for use in LMIC settings, for development of key themes. Transcripts will be revisited for accuracy and consistency by bilingual investigators raising data trustworthiness. Triangulating quantitative, qualitative and narrative data, findings across settings and within and between countries will be compared. The aim of this is to identify common themes for CST provision as well as geographical and cultural variations.

## Economic analysis

We will collect data on direct and indirect costs including the cost of delivering CST in each setting, use of services by people with dementia and caregivers, and on the time spent by caregivers in supporting people with dementia. We will attach country-specific unit costs to services. Incremental cost-effectiveness ratios will be computed, based on the (uncontrolled) pre–post design. This analysis will be conducted across the three countries by a team at the London School of Economics and Political Science.

### Phase IV: pathways to practice (months 29–36)

The aim of phase IV is to establish a model of good practice and a scalable plan, outlining ongoing and sustainable CST provision. We will engage with policymakers including utilising support obtained from Alzheimer Disease International (ADI), as part of a symposium at their international conferences. We will also examine key outcomes from the study of CST in phase III, in order to support the translation of CST into clinical practice for each country.

### Ensuring ongoing recruitment to CST groups

Through examining patterns of refusal, attendance, attrition and experience of CST from qualitative interviews, we will consider ways to counter this including psychoeducation to reduce stigma or providing increased support for transport. We will consider whether the inclusion/exclusion criteria were appropriate and adapt them if

needed. We will also create a sustainable system for CST groups following the completion of this research. This will require involvement of ongoing networks and financial agreements.

### Dissemination

We will examine the number of people trained, the quantum of required support and recommend sustainable training models including 'train the trainer' concepts. We will engage with universities and other course providers with the view of introducing CST into professional and vocational courses such as nursing, occupational therapy (OT), psychology, geriatric counsellors and care providers. This has been successful in Tanzania where CST is now taught routinely to undergraduate OT students at Kilimanjaro Christian Medical College. By the end of this study, we aim to have several people trained as CST trainers in different regions in each country, with clear plans about cascading this nationally. In remote communities in Tanzania, we will train primary healthcare workers with support and training from OT, so that groups can take place during the rainy season when access may be impossible for health workers from local towns.

We will refine and publish the adapted CST training manuals for each country. The availability of the manual and training infrastructure will be tailored for the needs and resources of each site. However, the rights to the CST manual in each country have been secured from the publishers.

Results of each phase of the trial will be disseminated at international conferences. Support for the current grant has been obtained by ADI, who will provide ongoing support including dissemination through newsletters and through a symposium at an ADI conference. A series of articles from each phase will also be submitted to high impact, peer-reviewed journals. Priority will be given to journals that specialise in research in lower/middle-income countries. Papers will include a systematic review of psychological and social interventions in LMICs, a report on the barriers to and facilitators of implementation in the different countries, results from the CST study and the qualitative interviews.

### Recommendation for routine outcome measures

To ensure that any beneficial effect of CST is documented, the psychometric properties of the measures will be assessed, including the ability to detect change and whether this change is supported by qualitative data gathered. We will assess whether they were appropriate for the setting and population to inform outcomes recommended for routine practice.

### Costs/affordability of models

We will: (1) examine the cost and potential affordability of CST; (2) calculate the total costs of supporting people with dementia pre-CST and post-CST; (3) investigate the cost-effectiveness of CST in each country from societal and health system perspectives and (4) appraise

the affordability of CST in consultation with local stakeholders.

## Data management plan

Data will be collected and analysed by researchers in each site, adhering to relevant data protection legislation for each country. All researchers have previously been trained for data handling and all electronic data will be stored on secure servers with inbuilt data encryption, automated back-up and anti-virus protection. Hard copies of data will be kept securely, in a locked cabinet and will be stored according to each institutions record retention policy. All data will be pseudonymised using a participant key and, to ensure that they comply with UK data protection legislation, UK sites will never have access to this key, rendering data anonymised. Anonymised data will be kept in line with University College London's (UCL) record retention policy.

## Study management and co-ordination

CST-International is sponsored by UCL. The programme manager (CS) from UCL will co-ordinate all work between three countries, support the researchers at each site, co-ordinate network meetings, assist with analysis of data and oversee any publications. Local teams will have monthly meetings with CS and AS using software such as Skype where tasks for the coming month will be discussed and decided on, with clear action points for each member of staff. Any operational difficulties will be discussed during team meetings, with advice sought from the appropriate institution where necessary. Representatives from each country will also meet annually in person to facilitate peer learning across countries.

An independent Advisory Group has been established and will meet once a year. Members include a representative from ADI, experts in CST, epidemiologists specialising in LMICs, a person living with dementia and a carer. The Advisory Group will advise the programme manager on all aspects of the research programme including methodology, dissemination and public engagement.

## Ethical considerations and approval

People with dementia will be included in this programme of work, both as PPI representatives and as participants in phase III. No personal information will be sought from PPI representatives. For phase III, people with dementia will be required to provide informed consent. To ensure capacity to consent, established screening procedures[20 31] in each country will be followed. CST has no documented adverse effects and, therefore, risk to participants will be low. Consent will be treated as an ongoing process and reaffirmation of consent will be sought at each participant contact. All participants and caregivers where appropriate will be provided with accessible information regarding phase III of the work and will be required to sign a consent form if they take part. Should participants be unable to write, thumb prints will be taken instead. Identifiable information will be held securely and

separately from research data in each country and will be deleted following completion of the project. Anonymised data will be transferred to the UK for analysis.

## Patient and public involvement

PPI will form an integral part of the research and representatives will be involved from phase I through to phase IV. As part of phase I, PPI representatives will be enlisted to codevelop implementation strategies for CST and, as part of phase IV, will help to implement these strategies across the three countries.

### Author affiliations

[1]Research Department of Clinical, Educational and Health Psychology, University College London (UCL), London, UK
[2]Department of Psychiatry, Centre of Excellence in Mental Health, Postgraduate Institute of Medical Education and Research (PGIMER) and Dr Ram Manohar Lohia Hospital, New Delhi, India
[3]Dementia Care, Schizophrenia Research Foundation (SCARF), Chennai, India
[4]Department of Research, Foundation for Research and Advocacy in Mental Health (FRAMe), Mysore, India
[5]Personal Social Services Research Unit (PSSRU), London School of Economics and Political Science (LSE), London, UK
[6]North Tyneside General Hospital, Northumbria Healthcare NHS Foundation Trust, North Shields, UK
[7]Institute for Health and Society, Newcastle University, Newcastle upon Tyne, UK
[8]Postgraduate Program of the Psychobiology Department, Universidade Federal de Sao Paulo, São Paulo, Brazil
[9]Institute of Psychiatry, Universidade Federal do Rio de Janeiro, Rio de Janeiro, Brazil
[10]Department of Psychology, PUC-Rio, Rio de Janeiro, Brazil
[11]Institute of Psychiatry, King's College London, London, UK
[12]Institute of Mental Health, University of Nottingham, Nottingham, UK
[13]Institute of Neuroscience, Newcastle University, Newcastle upon Tyne, UK
[14]Department of Psychiatry, Government Medical College, Kerala, India
[15]Department of Research, Schizophrenia Research Fondation (SCARF), Chennai, India

**Contributors** AS conceived the research, was primarily responsible for writing the protocol and acts as Chief Investigator for CST-International. CRS assisted in the writing of the protocol, costed the grant and prepared this manuscript for submission. CRS is the Programme Manager for CST-International. MC, SV, MKr, SKS and TR are the India coapplicants and researchers who assisted in developing the methodology for the proposal and writing the protocol. BD from the Mysore site contributed to the protocol. DCM, JL and CF are the Brazil leads and assisted in the writing of the protocol. RW, S-MP and CD are the Tanzania leads and assisted with the writing of the protocol. MWO and SM are UK based coapplicants, who commented on the protocol. AC-H and MKn provided information for the economic analysis of CST.

**Funding** This work is supported by the following Global Alliance for Chronic Diseases (GACD) funding agencies: The United Kingdom Medical Research Council (MRC: MR/S004009/1) and the Indian Council of Medical Research (ICMR: Indo-foreign/67/M/2018-NCD-I). No funding bodies were involved in the design, collection, analysis, interpretation or writing of the research or manuscript.

**Disclaimer** The views expressed are those of the authors and not necessarily those of GACD, the MRC or ICMR.

**Competing interests** None declared.

**Patient consent for publication** Not required.

**Ethics approval** Ethical approval was granted by the relevant body in each country. In Brazil, an ethics amendment was granted by the Federal University of Rio de Janeiro Institute of Psychiatry REC (ref: 57019616.5.0000.5263) to incorporate the current research programme. In India, approval was granted by Institutional Ethics Committees (IECs) in each of the four sites (Schizophrenia Research Foundation; SCARF: Chennai; 28 November 2017, Government Medical College: Thrissur; B6-8772/2016/MCTCR, PGIMER Dr RML Hospital: New Delhi; 219(38/2017)/IEC/PGIMER/RMLH43, All India Institute of Speech and Hearing:

Mysuru; SH/Extramural/1/2017–18). In Tanzania, approval was granted by KCMC and nationally by the National Institute of Medical Research, Dar-es-Salaam.

**Provenance and peer review** Not commissioned; externally peer reviewed.

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
