## [Reviewer comments · BMJ Open]

ARTICLE DETAILS

TITLE (PROVISIONAL)	Mixed methods implementation research of Cognitive Stimulation Therapy (CST) for dementia in lower and middle-income countries: Study protocol for Brazil, India and Tanzania (CST-International).
AUTHORS	Spector, Aimee; Stoner, Charlotte; Chandra, Mina; Vaitheswaran, Sridhar; Du, Bharath; Comas-Herrera, Adelina; Dotchin, Catherine; Ferri, Cleusa; Knapp, Martin; Krishna, Murali; Laks, Jerson; Michie, Susan; Mograbi, Daniel C.; Orrell, Martin; Paddick, Stella-Maria; KS, Shaji; Rangawsamy, Thara; Walker, Richard

VERSION 1 – REVIEW

REVIEWER	Iracema Leroi Global Brain Health Institute, Trinity College Dublin Ireland
REVIEW RETURNED	16-Apr-2019

GENERAL COMMENTS	Thank you for the opportunity to comment on this interesting study of implementing CST in LMICs. I have a few comments that may help the authors improve their manuscript. The abstract read a bit awkwardly and is should be made clear at the start that this is an implementation research study, not a protocol describing effectiveness of an intervention. The explanation is clear only once one reads further into the introduction. It should state something like: 'This is a protocol describing an implementation study that falls under the remit of the CST-International research programme, which aims to develop, test, refine and disseminate implementation strategies for CST for people with dementia in three diverse parts of the world'. I am very confused about the inclusion of a feasibility study with effectiveness outcomes in the middle of an implementation study. The background section cited how CST should be implemented in LMICs' ('Global adaptation and evaluation of CST coincided with the World Alzheimer's Report 2011, which stated that CST should routinely be given to people with early stage dementia and advocated CST as an effective, low-cost intervention in developing countries. To ensure that CST provision is translated into routine clinical practice, implementation research that addresses practical issues such as referral pathways and barriers to successful provision is needed'). This implies effectiveness and feasibility (acceptability, tolerability etc.) have all been tested and the focus is on implementation (i.e. getting an effective intervention into practice). Indeed, CST is already taught in Kilimanjaro! Hence, why are the authors going back to step on and testing feasibility!? This
---

	implies that next step should be a full scale RCT!. This makes no sense and needs clarification. Since the three countries selected are diverse (and on different continents!) it would be helpful to readers if a few lines of facts are listed about each one – population size, health and social care provision, status of dementia care etc. Otherwise, it seems like convenience sampling of sites. The inclusion of PPI is welcome, but the description makes it assumes that PPI is understood and routinely consulted in these countries. I really doubt that this is the case. Even in European countries this is most definitely not the case. The literature on PPI for dementia in LMICs is nearly non-existent. In my experience, consulting PPI members about study issues before they are fully aware of what is expected of them is an example of ‘research waste’. This is particularly the case in LMICs where the gulf between professional and patient is wide and patients often don’t expect to be consulted about their views and wishes. PPI should ideally be described as part of the training and capacity building first - that is, develop and train your PPI (researchers and the PPI groups) and only then consult them for your project. It seems implausible to involve PPI without raising awareness, knowledge and skills about PPI first. The protocol involves multiple steps are hard to follow. Using clear sub-headings and paragraphs for each key step would make this much clearer. A flow chart of all the steps would be helpful. Several different data types using different methodologies will be used to get the findings. A clearer theoretical basis for synthesizing the mixed method data should be employed rather than simply a descriptive approach. How will the findings from different data types be weighed or prioritized? How will conflicting or contradictory findings be managed? The authors might consult Brannen et al., for guidance on a suitable approach. In Phase one, ratings of aspects of implementation will be applied. What types of ratings? Have these been validated? How will they be analysed and synthesized with the findings from the other sites? Much more detail is needed here. In Phase One, regarding the plan for ‘The resulting matrix of ratings will be discussed within teams and decisions on which mechanisms to use will be made for subsequent implementation’ – how will these decisions be made? By consensus? By seniority? These are important issues, particularly since co-working in some LMICs may be challenging due to hierarchical professional structures. In other words, the opinion of the most senior team member may be deferred to. In Phase One, the ‘country specific framework’ is not clear – a clearer structure of the elements or domains that will be considered in the framework should be laid out. The content of this paragraph would be much clearer if put in tabular form with some coherence to the points raised. The adaptation of the CST manual for cultural appropriateness should be a separate step and not simply stuffed into the feasibility study as a simple task (which is how it reads now). It should
--	--

	involve consultation with the PPI group (assuming they have been properly trained) and some degree of field testing. Appropriate cultural adaptation of intervention tools is a huge task and not a simple matter of substituting a few items. Ideally, this adaptation will already have taken place in previous studies and the investigators can lean on this. What is the basis for the n=50 per site sample size? Of course a study such as this (assuming a stronger case has been made for the inclusion of a feasibility study, cf comments above) won't be a fully powered RCT, but sample size justifications still need to be included. Much more detail is required to make a convincing case for inclusion of the qualitative interviews. What is the anticipated sample size for the qualitative interviews? How will the researchers know that data saturation has been reached? Are team members trained? How will the interview schedule be developed? How will the cross-language issues be managed, or will the initial coding be done at each site? There is a literature of how this type of cross-national qualitative work should be done (i.e. cf Himmelsbach et al). The health economic measures need to be discussed in more detail. While those included are the standard ones used internationally, have they been adapted to the local settings? The questions on these scales need to be tailored to local circumstances and researchers need to be trained to extract the correct information. What health economic/cost effectiveness tariffs (country-specific unit costs to services) will be used? Do they exist for Tanzania? Which tariffs will be used for India and Brazil? These should be cited to be convincing to the reader. DMP: Data security, quality control and transfer of data is not mentioned and is a huge issue in doing this type of cross-national research in LMICs. A clear data management should be included, with a focus on data security, training of researchers in data handling etc. A clear DMP is essential to ensure credibility of the outcomes and protection of participants. Study coordination: Where will this take place and how will it be executed? Much detail is needed here to explain this to the reader. This is a complex study with different health systems in different time zones and involving researchers with different levels of experience. How will the teams communicate? By Zoom? Skype? Explain. The study management structure needs to be clearly outlined. Is there a Study Steering Committee with external members? What about the study management team – how will this be managed? Who is sponsoring the study? I doubt a single country will take sponsorship responsibility for the others. How is this managed? What is the mechanism for trouble shooting should operational problems arise? A discussion of the limitations and challenges of the research should be included. This is not 'research as usual'; there are significant challenges to consider.
--	--

REVIEWER	Areti Efthymiou
----------	-----------------

	Cyprus University of Technology, Cyprus
REVIEW RETURNED	28-Apr-2019

GENERAL COMMENTS	The development and implementation of CST program in Brazil, India and Tanzania are very important action that will assist people with dementia and their families. This protocol is interesting, multi-phase and ambitious. Overall, the protocol is well written Major comments The introduction needs more information on the role of non pharmacological therapies in Dementia and a stronger connection with the current status in these 3 countries. It would be important to add information on the current status of Non pharmacological therapies in these 3 countries: are there facilities providing this type of services: dementia centres etc. In Outcomes, even if there is information on the measure before and after the intervention, it might be important to add information on intervention characteristics when measuring, eg the number of sessions that each participant attended in the end of the intervention or to measure the quality of attendance per participant (by measuring time of interaction per participant if that is possible). Minor comments  1) Abstract: Even if it is a protocol, the abstract doesn't provide conclusions or expected results and ends abruptly. It would facilitate reader to add expected results or / and conclusions 2) Introduction, 1st paragraph line 14 and 15: erase brackets 3) Estimation of phase duration is missing, during which period you will deliver each phase? 4) Are there any foreseen limitations concerning the whole process and how do you consider to treat (risk management) 5) Page. 4 sentence 55 -58 , you might need to transfer this part to the method and include here more general information on CST 6) Phase One: we need more information in this first part of stakeholders meeting (who will be the stakeholder (estimated number), from all countries, who will moderate, how long they will last, how you will analyse the results, how you will integrate in next phase. You report that data will be organised into a country specific framework for CST implementation base on what theory ? 7) Page 8, sentence 47 " across a range of settings" please elaborate on the settings 8) Page 10 sentence 47: you are reporting 50 people per country: how this number has been calculated? What will be the process for recruiting, what channels will be used. There are 3 different countries with different cultures, here we need more details on the process. 9) Page. 10. Sentence 51: We would need more information on recruitment (where will take place the screening and for how long?), you should include validity and reliability information on the specific scales selected to assess dementia, are validated in the language of choice for these countries? 10) Page 10 sentence 10, please provide more information on the course and the aim of this in this phase
---

VERSION 1 – AUTHOR RESPONSE

Reviewer 1 - Revision

Thank you for the opportunity to comment on this interesting study of implementing CST in LMICs. I have a few comments that may help the authors improve their manuscript.

Response: We thank the reviewer for their comments and have amended the manuscript accordingly.

1) The abstract read a bit awkwardly and it should be made clear at the start that this is an implementation research study, not a protocol describing effectiveness of an intervention. The explanation is clear only once one reads further into the introduction. It should state something like: 'This is a protocol describing an implementation study that falls under the remit of the CST-International research programme, which aims to develop, test, refine and disseminate implementation strategies for CST for people with dementia in three diverse parts of the world'.

Response: We have amended the introduction section of the abstract to make it clear that this is an implementation study protocol e.g. 'This protocol describes CST-International; an implementation research study of CST'. We have also included the reviewer's suggestion of adding the aims to the abstract.

2) I am very confused about the inclusion of a feasibility study with effectiveness outcomes in the middle of an implementation study. The background section cited how CST should be implemented in LMICs ('Global adaptation and evaluation of CST coincided with the World Alzheimer's Report 2011, which stated that CST should routinely be given to people with early stage dementia and advocated CST as an effective, low-cost intervention in developing countries. To ensure that CST provision is translated into routine clinical practice, implementation research that addresses practical issues such as referral pathways and barriers to successful provision is needed'). This implies effectiveness and feasibility (acceptability, tolerability etc.) have all been tested and the focus is on implementation (i.e. getting an effective intervention into practice). Indeed, CST is already taught in Kilimanjaro! Hence, why are the authors going back to step on and testing feasibility!? This implies that next step should be a full scale RCT!. This makes no sense and needs clarification.

Response: Effectiveness and feasibility have indeed been previously explored. However, feasibility and acceptability outcomes previously established do not account for any implementation strategies we will develop and use over the course of the grant. For example, if one of our strategies is to target family carers with educational programmes regarding dementia and CST, this may impact on their willingness to engage with CST provision. As such the effectiveness and feasibility assessments mean that we are able to evaluate the effectiveness of our implementation strategies. When we discuss feasibility, we are referring to the feasibility of our implementation plans for CST rather than CST itself.

We understand how using this term can be confusing and we have removed 'feasibility' from the title of this section. We have also added the following clarification to the Methods and Analysis section of the abstract (p2) '3) Evaluation of implementation plans using a study of CST'.

Furthermore, we have added the following description to Phase Three (p8): 'The aims of Phase Three are to evaluate the feasibility of implementation strategies for each country. This will be accomplished using a study of CST where the following outcomes will be assessed...'

Regarding the use of outcome measures, this represents good practice and, in the UK and elsewhere, facilitators are encouraged to consistently monitor outcomes when delivering CST.

3) Since the three countries selected are diverse (and on different continents!) it would be helpful to readers if a few lines of facts are listed about each one – population size, health and social care provision, status of dementia care etc. Otherwise, it seems like convenience sampling of sites.

Response: We have included information regarding the prevalence of dementia in and the status of dementia care in each country (p4). We have also previously detailed why each country was selected on p6 (all completed pilot work, adapting the manual and engaged with local stakeholders). We are concerned that adding further demographic information about the countries will make the paper very lengthy. We are however happy to include this if the editor deems it appropriate.

4) The inclusion of PPI is welcome, but the description makes it assume that PPI is understood and routinely consulted in these countries. I really doubt that this is the case. Even in European countries this is most definitely not the case. The literature on PPI for dementia in LMICs is nearly non-existent. In my experience, consulting PPI members about study issues before they are fully aware of what is expected of them is an example of 'research waste'. This is particularly the case in LMICs where the gulf between professional and patient is wide and patients often don't expect to be consulted about their views and wishes. PPI should ideally be described as part of the training and capacity building first - that is, develop and train your PPI (researchers and the PPI groups) and only then consult them for your project. It seems implausible to involve PPI without raising awareness, knowledge and skills about PPI first.

Response: Stakeholders include people with dementia, carers, healthcare professionals, policy/decision makers and managers. All have previously been engaged and contributed to the adapted CST manual successfully in all three countries. We will not be consulting PPI members about 'study issues' but rather exploring what helps or hinders CST provision. To illustrate this, we have included an example question for each of the three groups (p7). We do not believe that it is appropriate or necessary to train people to respond to these questions as it may undermine their status as 'experts by experience'. However, whilst not undertaking formal training on PPI, we will preface stakeholder meetings with talks on both dementia, CST and what to expect from stakeholder groups (p7).

5) The protocol involves multiple steps are hard to follow. Using clear sub-headings and paragraphs for each key step would make this much clearer. A flow chart of all the steps would be helpful.

Response: Thank you for this suggestion. We have included subheadings for each of the key steps throughout the methodology.

6) Several different data types using different methodologies will be used to get the findings. A clearer theoretical basis for synthesizing the mixed method data should be employed rather than simply a descriptive approach. How will the findings from different data types be weighed or prioritized? How will conflicting or contradictory findings be managed? The authors might consult Brannen et al., for guidance on a suitable approach.

Response: Thank you for this suggestion. We have reviewed Brannen 2007 and note that this approach is particularly relevant for how quantitative and qualitative data are combined. In this protocol, qualitative and quantitative data will be treated as distinct and we do not propose to weigh or prioritise according to data type. It is unlikely that data will be contradictory. We are using implementation frameworks to guide an implementation process and testing the success of this using a CST study.

7) In Phase one, ratings of aspects of implementation will be applied. What types of ratings? Have these been validated? How will they be analysed and synthesized with the findings from the other sites? Much more detail is needed here.

Response: We think this refers to the bottom of page 7/ start of page 8, where we incorrectly used the word 'analysed' instead of tabulated. For reference, all implementation mechanisms will be tabulated alongside the barriers and facilitators they refer to. The rating system uses the mode to rate how essential and how easy each of the mechanisms are to use. We have amended the information under the subheading 'developing implementation mechanisms' to provide more clarity on this procedure. It is important to reiterate that this is an implementation research study, for which no standardised methodology exists. We have used an existing theoretical model of implementation to develop a novel methodology that can be employed in most contexts.

8) In Phase One, regarding the plan for 'The resulting matrix of ratings will be discussed within teams and decisions on which mechanisms to use will be made for subsequent implementation' – how will these decisions be made? By consensus? By seniority? These are important issues, particularly since co-working in some LMICs may be challenging due to hierarchical professional structures. In other words, the opinion of the most senior team member may be deferred to.

Response: We are asking the teams to reach a consensus and to justify which mechanisms they use in a written report. This is then reviewed and approved by the UK team. The written report will help to negate the seniority effect. We have provided clarification on this (p8 'Agreeing an 'Implementation Plan').

9) In Phase One, the 'country specific framework' is not clear – a clearer structure of the elements or domains that will be considered in the framework should be laid out. The content of this paragraph would be much clearer if put in tabular form with some coherence to the points raised.

Response: We think this refers to Phase Two rather than Phase One and agree that this could be clearer. We have amended this information under 'agreeing an implementation plan' p8.

10) The adaptation of the CST manual for cultural appropriateness should be a separate step and not simply stuffed into the feasibility study as a simple task (which is how it reads now). It should involve consultation with the PPI group (assuming they have been properly trained) and some degree of field testing. Appropriate cultural adaptation of intervention tools is a huge task and not a simple matter of substituting a few items. Ideally, this adaptation will already have taken place in previous studies and the investigators can lean on this.

Response: As stated in the introduction (bottom of pg5 - top of pg6), the manual has previously been adapted for these countries using established guidelines which includes PPI. We think the reviewer is referring to the example of adaptations given at the bottom of p10. We have reiterated that all manuals were previously adapted using an established procedure.

11) What is the basis for the n=50 per site sample size? Of course a study such as this (assuming a stronger case has been made for the inclusion of a feasibility study, cf comments above) won't be a fully powered RCT, but sample size justifications still need to be included.

Response: The reviewer is correct in that that this is not a RCT and therefore sample size calculations were not performed. Our aim is to gather enough data to ascertain whether our implementation strategies across the three countries have been sufficient. As such, the sample size was calculated on a pragmatic basis, based on the number of groups needed in each setting to evaluate this, with

reference to the site's available time and resources. This will also give an indication of effect size. We have clarified this in the manuscript (p9).

12) Much more detail is required to make a convincing case for inclusion of the qualitative interviews. What is the anticipated sample size for the qualitative interviews? How will the researchers know that data saturation has been reached? Are team members trained? How will the interview schedule be developed? How will the cross-language issues be managed, or will the initial coding be done at each site? There is a literature of how this type of cross-national qualitative work should be done (i.e. cf Himmelsbach et al).

Response: We have added more clarity on the justification for using qualitative interviews and specified that there are trained qualitative researchers in each site who will be responsible for conducting this methodology (p12). We have previously specified that this will be developed using the CFIR (p11).

13) The health economic measures need to be discussed in more detail. While those included are the standard ones used internationally, have they been adapted to the local settings? The questions on these scales need to be tailored to local circumstances and researchers need to be trained to extract the correct information. What health economic/cost effectiveness tariffs (country-specific unit costs to services) will be used? Do they exist for Tanzania? Which tariffs will be used for India and Brazil? These should be cited to be convincing to the reader.

Response: We have added information regarding local adaptations to the CSRI and RUD (p10). Manuscripts detailing specific methodology for the health economic measures and results will be written separately so that they can be given the detail they deserve.

14) DMP: Data security, quality control and transfer of data is not mentioned and is a huge issue in doing this type of cross-national research in LMICs. A clear data management should be included, with a focus on data security, training of researchers in data handling etc. A clear DMP is essential to ensure credibility of the outcomes and protection of participants.

Response: We have added a new subheading detailing our data management plan (p16).

15) Study coordination: Where will this take place and how will it be executed? Much detail is needed here to explain this to the reader. This is a complex study with different health systems in different time zones and involving researchers with different levels of experience. How will the teams communicate? By Zoom? Skype? Explain.

Response: We have added a new subheading detailing the study management and co-ordination (p18)

16) The study management structure needs to be clearly outlined. Is there a Study Steering Committee with external members? What about the study management team – how will this be managed? Who is sponsoring the study? I doubt a single country will take sponsorship responsibility for the others. How is this managed? What is the mechanism for trouble shooting should operational problems arise?

Response: We have added details regarding the study management, advisory group and sponsorship to the protocol (p18)

17) A discussion of the limitations and challenges of the research should be included. This is not 'research as usual'; there are significant challenges to consider.

Response: We are aware that challenges will arise as part of this research and will monitor this as the work goes on. However, we have followed the guidelines for submitting protocols to BMJ Open closely and note that they do not require a discussion section. We have, therefore, chosen to include these in later publications.

Reviewer: 2

Reviewer Name: Areti Efthymiou

Institution and Country: Cyprus University of Technology, Cyprus

Please state any competing interests or state 'None declared': none declared

Please leave your comments for the authors below

The development and implementation of CST program in Brazil, India and Tanzania are very important action that will assist people with dementia and their families. This protocol is interesting, multi-phase and ambitious. Overall, the protocol is well written

Response: We thank the reviewer for his comments and have amended the manuscript accordingly.

Major comments

1) The introduction needs more information on the role of non pharmacological therapies in Dementia and a stronger connection with the current status in these 3 countries.

Response: We have added a paragraph to the introduction (p4) strengthening the rationale for non-pharmacological interventions.

2) It would be important to add information on the current status of Non pharmacological therapies in these 3 countries: are there facilities providing this type of services: dementia centres etc.

Response: We believe that we have referenced the fact that services are fragmented and non-existent in each of these countries (p4) and that further information would be beyond the scope of the current manuscript.

3) In Outcomes, even if there is information on the measure before and after the intervention, it might be important to add information on intervention characteristics when measuring, eg the number of sessions that each participant attended in the end of the intervention or to measure the quality of attendance per participant (by measuring time of interaction per participant if that is possible).

Response: We believe these are important outcomes and are covered in our 'feasibility analysis' on p15.

Minor comments

1) Abstract: Even if it is a protocol, the abstract doesn't provide conclusions or expected results and ends abruptly. It would facilitate reader to add expected results or / and conclusions

Response: We have followed the guidelines for submitting to protocols to BMJ and they do not allow conclusions/ results in an abstract.

2) Introduction, 1st paragraph line 14 and 15: erase brackets

Response: We have erased these brackets.

3) Estimation of phase duration is missing, during which period you will deliver each phase?

Response: We have now included phase duration details for each heading.

4) Are there any foreseen limitations concerning the whole process and how do you consider to treat (risk management)

Response: CST has no adverse effects and we do not foresee any risk to participants. We have now noted this in the protocol (p19).

5) Page. 4 sentence 55 -58 , you might need to transfer this part to the method and include here more general information on CST

Response: We think this refers to the background of CST given in the introduction (3rd paragraph). We believe that it is appropriate to keep this information in the introduction and, for specific CST methodology, the reader can use references provided.

6) Phase One: we need more information in this first part of stakeholders meeting (who will be the stakeholder (estimated number), from all countries, who will moderate, how long they will last, how you will analyse the results, how you will integrate in next phase. You report that data will be organised into a country specific framework for CST implementation base on what theory ?

Response: We have added more details on this process (pg 7).

7) Page 8, sentence 47 “ across a range of settings” please elaborate on the settings.

Response: We have elaborated on this (pg 9).

8) Page 10 sentence 47: you are reporting 50 people per country: how this number has been calculated? What will be the process for recruiting, what channels will be used. There are 3 different countries with different cultures, here we need more details on the process.

Response: Please see our response to Reviewer 1, point 11.

9) Page. 10. Sentence 51: We would need more information on recruitment (where will take place the screening and for how long?), you should include validity and reliability information on the specific scales selected to assess dementia, are validated in the language of choice for these countries?

Response: We have attempted to expand on this but, as this is an implementation study, referral pathways and strategies will be developed over the course of Phase 2. We have also expanded on our assertion that all measures have been validated in each country (p10).

10) Page 10 sentence 10, please provide more information on the course and the aim of this in this phase

Response: As with the above, we have expanded slightly but this course will be developed over Phase Two (p11).

VERSION 2 – REVIEW

REVIEWER	Areti Efthymiou Cyprus University of Technology, Cyprus
REVIEW RETURNED	04-Jul-2019

GENERAL COMMENTS	General comment: Very interesting and well written protocol. All aspects are covered, it is easy for the reader to understand the different phases. It is an ambitious project that will require very strict coordination. Major comments 1) It would be important to add in all phases if possible: "who will be the local staff (researcher, staff of services, health -related organisations public or private" and if you have already established this collaboration or this is also part of the activities you foresee, "where the different activities will take place (location)" . For example in phase 1: Who will be responsible for making the communication with the stakeholders (researchers, health care professionals, staff of ngos), which local services and stakeholders are involved as core partners (public, health services, ngos), where (predicted location) the focus groups and interventions will take place and how the researchers, policy makers etc will be invited. Alternatively, you may include in the methods section a separate paragraph introducing the phases including this information for who is involved locally, where the activities will be delivered and how you have established communication eg Study settings and local participation Minor comments Introduction 1) P. 4, 3rd paragraph. 2 first sentences "In countries.... Dementia" please support this statement by adding references in the end. Method 2) It might be helpful to add an annex with the selected questions for the focus groups. 3) p. 7 "staff in each country": staff from which organizations: public, ngo, universities, or any other service 4) p.8 Please add definition of the CST-international team 5) p.8 local coordinator: member of CST-international team and staff of a local association? Please provide a description. 6) p.9 "to maximise.... Day centres" How the communication will take place with the health care setting to participate in the study? 7) p.9 "required to evaluate the success of implementation strategies". Please include reference for this decision 8) p. 11 please include location (dementia centers, other health facilities, or cities) where the 14 sessions of CST will be delivered
---

VERSION 2 – AUTHOR RESPONSE

Authors Response to Reviewers

General comment:

Very interesting and well written protocol. All aspects are covered, it is easy for the reader to understand the different phases. It is an ambitious project that will require very strict coordination.

Response: We thank the reviewer for their positive feedback.

Major comments

- 1) It would be important to add in all phases if possible: "who will be the local staff (researcher, staff of services, health -related organisations public or private" and if you have already established this collaboration or this is also part of the activities you foresee, "where the different activities will take place (location)" . For example in phase 1: Who will be responsible for making the communication with the stakeholders (researchers, health care professionals, staff of ngos), which local services and stakeholders are involved as core partners (public, health services, ngos), where (predicted location) the focus groups and interventions will take place and how the researchers, policy makers etc will be invited. Alternatively, you may include in the methods section a separate paragraph introducing the phases including this information for who is involved locally, where the activities will be delivered and how you have established communication eg Study settings and local participation

Response: Thank you for this suggestion. We have added an introductory paragraph to the methods and analysis section detailing the national and international teams. We have also specified the setting in which the teams are employed e.g. higher education, NGOs etc (p6). We have also added a clarification to p7 specifying that national teams will contact stakeholders.

Minor comments

Introduction

- 1) P. 4, 3rd paragraph. 2 first sentences "In countries.... Dementia" please support this statement by adding references in the end.

Response: We have provided a reference from the 2016 World Alzheimer Report by Alzheimer's Disease International (ADI) to support this statement.

Method

- 1) It might be helpful to add an annex with the selected questions for the focus groups.

Response: We have previously included example questions from each stakeholder group (p7) but it is important to reiterate that these are not focus groups but stakeholder meetings. The stakeholder meetings will be addressed in sufficient detail in future publications.

- 2) p. 7 "staff in each country": staff from which organizations: public, ngo, universities, or any other service.

Response: Please see our response to Major Comments, Point 1.

- 3) p.8 Please add definition of the CST-international team

Response: We have added an explanation of what we mean by the CST-International team to the first paragraph of the methods section (p6).

- 4) p.8 local coordinator: member of CST-international team and staff of a local association?
Please provide a description.

Response: We have added a brief description of the local co-ordinator role (p8).

- 5) p.9 "to maximise.... Day centres" How the communication will take place with the health care setting to participate in the study?

Response: This will be unique to each setting and we have specified that contact should be made using the most appropriate means. We have given two examples of both email contact and a formal letter. We have also specified that teams should meet with managers of sites prior to their recruitment (p10).

- 6) p.9 "required to evaluate the success of implementation strategies". Please include reference for this decision

Response: As we have noted, the sample size was calculated on a pragmatic basis based on our discussions in national and international teams regarding their available time and resources. As such, it is not possible to give a reference for these discussions.

- 7) p. 11 please include location (dementia centers, other health facilities, or cities) where the 14 sessions of CST will be delivered

Response: As stated on the bottom of p11, we have noted that recruitment will take place in 'Rio de Janeiro and Sao Paulo in Brazil; Chennai, Mysuru, Kerala and New Delhi in India; Kilimanjaro and Arusha in Tanzania. Groups will take place in both rural and urban settings which include outpatient units, carers associations, Primary Health Care Centres and community settings'. It is important to note that, as this is the protocol for an implementation study, specific venues are likely to be recruited using our implementation strategies as part of Phase 2. We have also added this clarification to p11.